

# Integrative species delimitation in the common ophiuroid *Ophiothrix angulata* (Echinodermata: Ophiuroidea): insights from COI, ITS2, arm coloration, and geometric morphometrics

Yoalli Quetzalli Hernández-Díaz[1,2,3], Francisco Solis[2], Rosa G. Beltrán-López[4,5], Hugo A. Benítez[6,7], Píndaro Díaz-Jaimes[8] and Gustav Paulay[9]

[1] Posgrado en Ciencias del Mar y Limnología, Universidad Nacional Autónoma de México, Ciudad de México, México

[2] Laboratorio de Sistemática y Ecología de Equinodermos, Instituto de Ciencias del Mar y Limnología, Universidad Nacional Autónoma de México, Ciudad de México, México

[3] Unidad Multidisciplinaria de Docencia e Investigación - Sisal, Facultad de Ciencias, Universidad Nacional Autónoma de México, Yucatán, México

[4] Laboratorio de Ictiología, Centro de Investigaciones Biológicas, Universidad Autónoma del Estado de Morelos, Cuernavaca, Morelos, México

[5] Departamento de Zoología, Instituto de Biología, Universidad Nacional Autónoma de México, Ciudad de México, México

[6] Laboratorio de Ecología y Morfometría Evolutiva, Centro de Investigación de Estudios Avanzados del Maule, Instituto Milenio Biodiversidad de Ecosistemas Antárticos y Subantárticos (BASE), Universidad Católica del Maule, Talca, Chile

[7] Centro de Investigación en Recursos Naturales y Sustentabilidad (CIRENYS), Universidad Bernardo O'Higgins, Santiago, Chile

[8] Unidad Académica de Ecología y Biodiversidad Acuática, Instituto de Ciencias del Mar y Limnología, Universidad Nacional Autónoma de México, Ciudad de México, México

[9] Florida Natural History Museum, University of Florida, Gainesville, FL, United States of America

Corresponding author
Yoalli Quetzalli Hernández-Díaz, quetzalli.hernandez@ciencias.unam.mx, quetzalli.hernandez@gmail.com

## ABSTRACT

*Ophiothrix angulata* (Say, 1825) is one of the most common and well-known ophiuroids in the Western Atlantic, with a wide geographic and bathymetric range. The taxonomy of this species has been controversial for a century because of its high morphological variability. Here we integrate information from DNA sequence data, color patterns, and geometric morphometrics to assess species delimitation and geographic differentiation in *O. angulata*. We found three deeply divergent mtDNA-COI clades (K2P 17.0–27.9%). ITS2 nuclear gene and geometric morphometrics of dorsal and ventral arm plates differentiate one of these lineages, as do integrative species delineation analyses, making this a confirmed candidate species.

## INTRODUCTION

Members of the Class Ophiuroidea, brittle stars, basket stars, and snake stars, are the most diverse of the five echinoderm classes, and are often abundant and, at times, dominant

members of marine benthic communities (*Kissling & Taylor, 1977*; *Lewis & Bray, 1983*; *Hendler & Littman, 1986*; *Aronson, 1988*; *Stöhr, O'Hara & Thuy, 2012*; *Boissin et al., 2016*). While the taxonomy of ophiuroids was thought to be well known, sequence data has demonstrated that numerous well-known brittle stars are complexes of cryptic species (*e.g.*, the cosmopolitan *Amphipholis squamata* (*Boissin, Féral & Chenuil, 2008*) and the Northeastern Atlantic *Ophiothrix fragilis* (*Pérez-Portela, Almada & Turon, 2013*; *Taboada & Pérez-Portela, 2016*)).

The tropical Western Atlantic (WA) is home to the second major coral reef realm, has very high diversity and endemicity, and hosts a remarkably abundant, rich, and ecologically diverse brittle star fauna (*Kissling & Taylor, 1977*; *Stöhr, O'Hara & Thuy, 2012*; *Sobha, Vibija & Fahima, 2023*). The Ophiotrichidae (*Ljungman, 1867*) is the third-largest family of brittle stars and is well-represented in the WA. Ophiuroids have undergone extensive systematic revision following recent phylogenomic studies, and ophiotrichids were found to be "genetically and morphologically coherent" (*O'Hara et al., 2018*). Although ophiuroids have become well-defined at the family level, substantial work remains on many genera and species. *Ophiothrix* is the largest brittle star genus with 96 accepted species (*Dos Santos Alitto et al., 2019*; *Santana et al., 2020*; *Stöhr, O'Hara & Thuy, 2023*), but neither the genus nor its subgenera are monophyletic (*O'Hara et al., 2017*; *O'Hara et al., 2018*). The high diversity and morphological variability of ophiotrichids have made their species-level taxonomy challenging (*Clark, 1946*; *Clark, 1967*; *Tommasi, 1970*; *Hoggett, 1991*; *Hendler, 2005*).

Traditional species delineation of ophiuroids has focused on macro-morphological characters that can show substantial variability and overlap among species, causing taxonomic uncertainty (*Arlyza et al., 2013*). An integrative approach that combines information from live appearance, microstructure, life history, ecology, ethology, physiology, distribution, and especially DNA sequence data provides information to resolve species in challenging groups like *Ophiothrix* (*Bickford et al., 2007*; *Boissin, Féral & Chenuil, 2008*; *Padial et al., 2010*; *Pérez-Portela, Almada & Turon, 2013*; *O'Hara et al., 2014a*; *Richards, De Biasse & Shivji, 2015*; *Taboada & Pérez-Portela, 2016*; *Dos Santos Alitto et al., 2019*; *Newton et al., 2020*). Over the past decade, the use of microstructural characters has emerged as a valuable tool in systematic studies of brittle stars, revealing their phylogenetic value (*O'Hara et al., 2014b*; *Thuy & Stöhr, 2016*). Among these characters, arm plates have proven to be particularly important in establishing a congruence with molecular data, enabling the inference of phylogenetic relationships even at the genus level (*Thuy & Stöhr, 2016*). Therefore, these characters offer a valuable approach to analyze species complexes and contribute to species delimitation.

*Ophiothrix angulata* (*Say, 1825*) was one of the first brittle stars and the first ophiotrichid described for the Americas. It is nearly ubiquitous along the Atlantic coasts of North and South America in warm temperate to tropical waters, from North Carolina, USA to at least Venezuela, and throughout the Caribbean islands, Bahamas, and Bermuda (*Devaney, 1974*; *Herrera-Moreno & Betancourt Fernández, 2004*; *Alvarado, Solís-Marín & Ahearn, 2008*; *Borrero-Pérez et al., 2008*; *Del Valle García et al., 2008*; *Laguarda-Figueras et al., 2009*; *Alvarado & Solís-Marín, 2013*; *Noriega & Fuentes-Carrero, 2014*; *Sandino et al., 2017*; *GBIF,*

*2022*). Records of *O. angulata* further south are now attributed to other species (*Santana et al., 2017*; *Santana et al., 2020*). The species also has a broad bathymetric distribution from intertidal to bathyal depths (~1,000 m; ICML-UNAM 3.34.40: 770 m; MCZ OPH-30910: 1,499 m). This species is capable of inhabiting corals, sponges, live under rocks, and among turf algae, which gives it a great capacity to adapt to different micro-habitats.

*Ophiothrix angulata* is highly variable and has a broad latitudinal and bathymetric distribution that has attracted taxonomic attention (*Tommasi, 1970*; *Clark, 1933*; *Hendler et al., 1995*; *Hendler et al., 1999*; *Hendler, 2005*; *Santana et al., 2017*). Variation is especially notable in color (*Lyman, 1865*; *Verrill, 1899*; *Clark, 1901*), and *Clark (1918)* named five varieties based on this. The species has also been noted to vary in disc shape, and arrangement of spinelets around the disc, but *Hendler et al. (1995)*, concluded that this variation does not seem to sort into species-level units.

The goal of this study was to assess whether the great morphological diversity of *O. angulata* is the result of high intra-specific variation or differentiation among multiple cryptic or pseudo-cryptic species. We tested species boundaries using an integrative taxonomic approach, by combining mtDNA COI and nrDNA ITS2 sequence data, color patterns, and geometric morphometrics of dorsal and ventral arm plates. We combined results from genetic and morphological assessments for species delimitation using an integrated Bayesian phylogenetic and phylogeographic approach (iBPP) (*Solís-Lemus, Knowles & Ané, 2015*). We also analyzed the population diversity and demographic history of the clades discovered.

## MATERIALS & METHODS

### Sampling sites and collections

We used 146 *Ophiothrix angulata* specimens from 24 localities across the West Atlantic (Fig. 1; Table S1). Thirty-five samples were collected specifically for this project; others were obtained from collections at the Invertebrate Zoology Collection, Florida Museum of Natural History, University of Florida (UF); University of West Florida (UWF); Natural History Museum of Los Angeles County (LACM); Colección Nacional de Equinodermos "Dra. María Elena Caso Muñoz", Instituto de Ciencias del Mar y Limnología, UNAM, México (ICML-UNAM); Colección Regional de Equinodermos de la Península de Yucatán, UMDI-Sisal, UNAM, México (COREPY-UNAM), and Museo de Zoología, Escuela de Biología, Universidad de Costa Rica (MZ-UCR). Field sample collection was approved by Secretaría de Agricultura, Ganadería, Desarrollo Rural, Pesca y Alimentación (SAGARPA: Permission number: PPF/DGOPA-082/19). All 146 individuals were sequenced for COI, 14 for ITS2, 46 were selected for geometrics morphometric analyses while living color pattern was examined for 46 specimens from the UF and COREPY photographic collections. Twelve additional sequences available from GenBank were also used (Table S1). Nine terminals from five outgroup species were also sequenced: *Ophiothrix cimar Hendler, 2005*; *Ophiothrix lineata Lyman, 1860*; *Ophiothrix stri Hendler, 2005*; *Ophiothrix suensonii Lütken, 1856* and *Ophiactis savignyi* (*Müller & Troschel, 1842*).

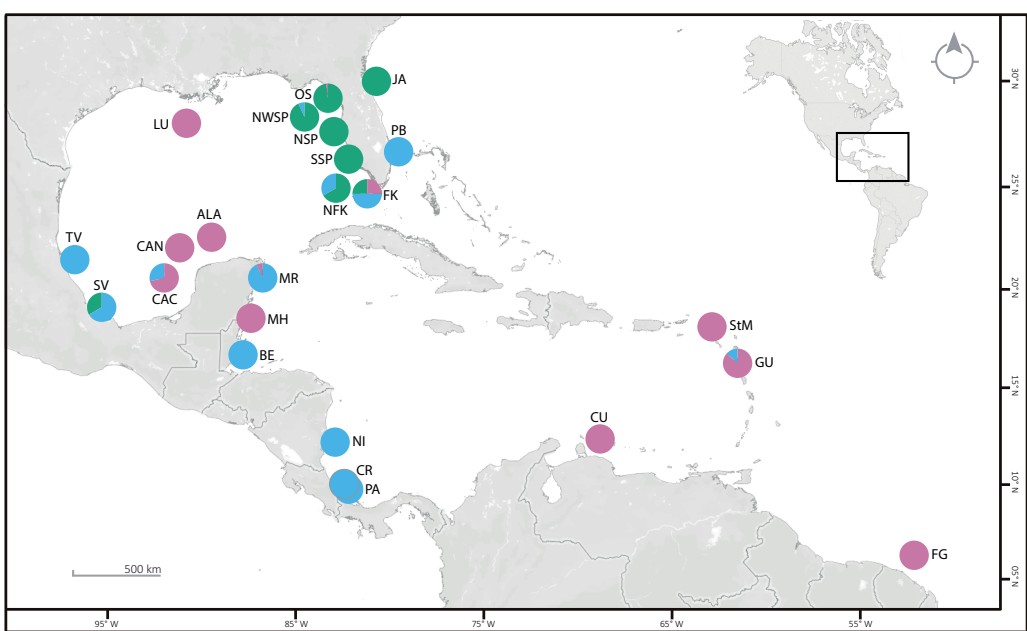

**Figure 1** **Sampling locations (see Table S1 for details), with the proportion of specimens per clade are given as pie charts.** Clade 1 (green), Clade 2 (blue), and Clade 3 (pink). OS: off Steinhatchee, North of St. Petersburg and Cedar Key. NWSP, Northwest of St. Petersburg; NSP and SSP, North and South of St. Petersburg; NFK, North of Florida Keys; FK, Florida Keys; PB, Palm Beach; JA, near Jacksonville; LU, near Louisiana; TV, Tuxpan Veracruz; SV, Sistema Arrecifal Veracruzano and Monte Pio; CAC, Cayo Arcas area; CAN, Cayo Arenas; ALA, Alacranes Reef; MR, Mayan Riviera; MH, Mahahual; BE, Belize; NI, Nicaragua; CR, Costa Rica; PA, Panama; StM, St. Martin; CU, Curaçao; GU, Guadeloupe; FG, French Guiana. ©OpenStreetMap 2022.

## DNA extraction and sequence alignment

DNA was extracted from ethanol-fixed arm tissue using the Chelex protocol (*Walsh, Metzger & Higuchi, 1991*) or the Omega Bio-Tek E.Z.N.A. Mollusc DNA kit according to the manufacturer's instructions. The echinoderm barcoding primers COIceF and COIceR (*Hoareau & Boissin, 2010*) were used to amplify a 655-base pair (bp) region of COI as described by *Michonneau & Paulay (2014)*. Electropherograms were checked, assembled into contigs, and manually edited using Sequencher 4.6 (Gene Code Corps, Ann Arbor, MI, USA). Consensus sequences were aligned using Muscle (*Edgar, 2004*), and alignment was verified by eye using PhyDE v.10.0 (*Müller et al., 2010*). The sequence alignment was converted to protein using Genius v8.1.7 (*Kearse et al., 2012*) to ensure a proper reading frame and to verify the absence of stop codons. A 527 pb section of the nuclear DNA internal transcribed spacer-2 (ITS2) was amplified using the primers OphITS2F and OphITS2R as described by *Naughton et al. (2014)*. Sequences were deposited in GenBank (COI accession numbers: MT338285–MT338398, ON245084–ON245096; ITS2 accession numbers: OQ225473–OQ225482).

## Phylogenetic analyses

Phylogenetic analyses using Bayesian Inference (BI) and Maximum Likelihood (ML) methods were performed separately for COI and ITS2 sequence datasets on the CIPRES Science Gateway portal (*Miller, Pfeiffer & Schwartz, 2010*). jModelTest2 (*Darriba et al., 2012*) was used on the CIPRES portal to select the best model of molecular evolution based on Akaike information criteria tests (AIC). ML analyses were done using RAxML-HPC2 on XSEDE (*Stamatakis, 2006*; *Stamatakis, Hoover & Rougemont, 2008*) with the GTR+GAMMA model of sequence evolution; nodal support values were assessed with 1,000 rapid bootstraps (*Felsenstein, 1985*). BI analyses were performed using MrBayes v.3.2.7a on XSEDE (*Ronquist et al., 2012*), using the GTR+I+G model. The MCMC search was based on two independent runs of four chains each and 6,000,000 generations (sampled every 1,000 generations) until the final average and standard deviation were close to 0.01. Twenty-five percent of the initial trees were discarded as Burn-in. Results were summarized in Tracer v.1.7.1 (*Rambaut et al., 2018*) based on the Effective Sample Size-ESS for each parameter. The gene phylogenies were represented using FigTree v.1.4.4 (*Rambaut, 2018*) and annotated using Adobe Illustrator CC v.2017-22.0.1.

## Geometric morphometric analyses

We analyzed dorsal arm plates (DAP) and ventral arm plates (VAP) from 46 specimens. Images of DAP and VAP were obtained through scanning electron microscopy (SEM) from intermediate-sized specimens with disc diameters (DD) of 2.5 to 5.5 mm to assess the size-independent variability. The integument was removed from the 4th–8th arm segments (counted from the first arm segment that contained a regular vertebra and lateral plates *Stöhr, 2005*), due to the adult proximal arm plates showing the highest degree of morphological differentiation, reflecting differences between species (*Thuy & Stöhr, 2011*). The integument removal was performed by submergence in 0.3% sodium hypochlorite solution for 2–8 h, washed with distilled water and 98% ethanol, air-dried, and mounted on aluminum stubs using carbon tape. The samples were then gold-coated and scanned using a Hitachi-SU1510 SEM at the LANABIO facility at the Instituto de Biología, UNAM.

In order to organize the data for analyses, the file format of the plate images was imported into TPS file format using tpsUtil v.1.58. Landmarks and curves were digitized using tpsDig2 v.2.17 (*Rohlf, 2015*). Along the external margin of each DAP, five homologous type II landmarks and sixty-eight evenly-spaced semi-landmarks were digitized (Fig. 2A). The same procedure was followed for VAP but using four landmarks type II and 66 semi-landmarks evenly-spaced (Fig. 2B). Geometric morphometric (GM) analyses followed the outlined by *Masonick & Weirauch (2019)*, using the R package *geomorph* v.3.2.1 R package (*Adams & Otárola-Castillo, 2013*). The GM analyses compared four clades and subclades defined by COI sequence data set (Table S2): Clade 1A, Clade 1B, Clade 2A, and Clade 3. Generalized Procrustes analysis was used to extract the shape data for comparison, removal, translation, scaling, and rotation of all selected landmarks. A proxy of size in GM is centroid size which is the square root of the sum of squared distances of an object's landmarks from their centroid or center of gravity.

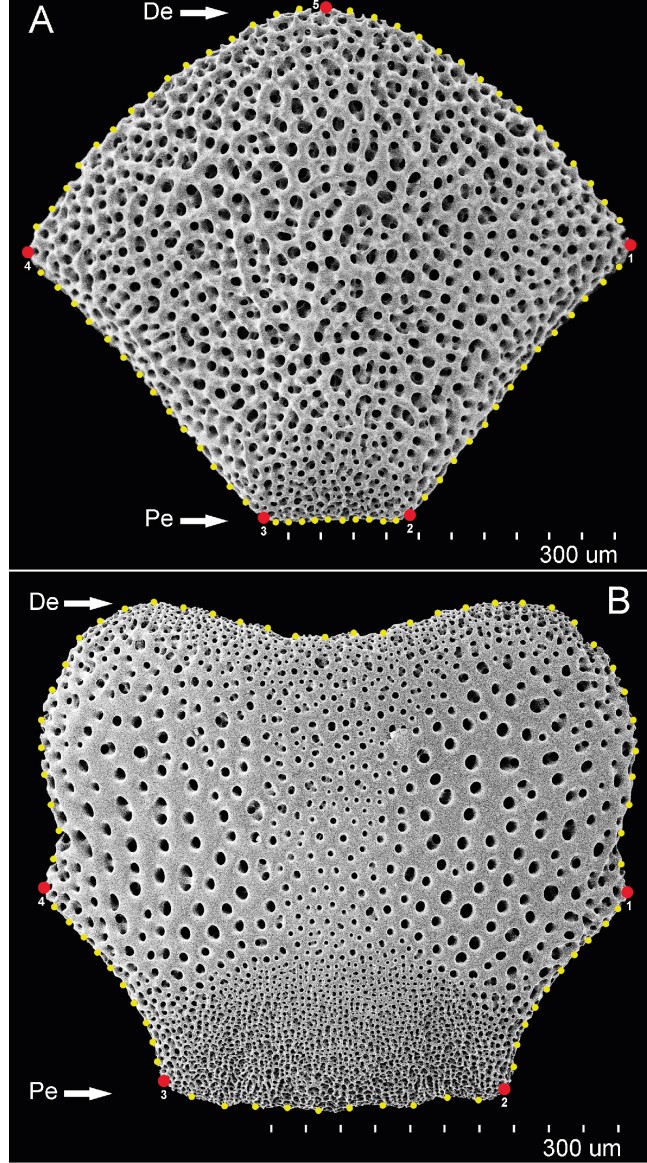

**Figure 2** **DAP and VAP landmarks.** Landmarks type II (red) and semi-landmarks (yellow) used for geometric morphometrics to investigate variation in arm plates in *Ophiothrix angulata*. Numbers indicate landmark position. (A) Dorsal arm plate (DAP). (B) Ventral arm plate (VAP). Pe, Proximal edge; De, Distal edge. Photos credit: Y. Quetzalli Hernández-Díaz.

The semi-landmarks digitized for each arm plate were optimized to reduce bending energy using the function ProcD = False in *geomorph*, thus providing the best fit during optimization. Shape variation was analyzed through principal component analysis (PCA) using the covariance matrix of the individual, and a graphical scatterplot was performed using the two principal components, which accumulated the maximum variance of the data (PC1 *vs.* PC2; S3 Appendix). Thin-plate spline deformation grids were calculated from the mean shape variance along each PC axis with the *shapes* v.1.2.5 R package (*Dryden, 2018*),

representing the overall shape modification. To enhance the visualization of data dispersion, two scatterplots in 3D were constructed based on the first three PCs using the R package *car* v.3.0-6 (*Fox, Weisberg & Price, 2018*; *Masonick & Weirauch, 2019*). A Procrustes analysis of variance (ANOVA) was performed on all analyses using the "procD.lm" function in *geomorph*. The resulting pairwise Procrustes distances were compared to assess the significance of differences in mean arm plate shape among the groups. The statistical significance of the observed variation was assessed through a permutation test of the randomized model residuals with 999 iterations at an $\alpha$-value of 0.05.

## Measurement error and allometry-test

To evaluate the impact of measurement error, the selected landmarks were digitized twice on each image (dorsal and ventral arm plates), on different days, by the same observer (*Viscosi & Cardini, 2011*). The error was calculated as percent measurement error (%ME) by comparing the variation among measurements based on a formula developed by (*Bailey & Byrnes, 1990*; *Yezerinac, Lougheed & Handford, 1992*). A Procrustes ANOVA on the residuals of Procrustes distances was used to compare within and among individual shape variance components. The allometry was evaluated by regression of the Procrustes shape coordinates on centroid size using a $\log_{10}$ scale. Interaction between centroid size and "clade" (dorsal or ventral arm plates grouped by clade) factor was estimated with Procrustes ANOVA in *geomorph* v.3.2.1 (*Klingenberg, 2016*).

## Dorsal arm color pattern analysis

Color patterns were examined from live-taken images associated with 46 sequenced specimens from the UF and COREPY invertebrate image collections (Table S1). Twenty-five color characters were selected from dorsal arm views (Figs. S1, S2, and S3). All characters were treated as discrete and unordered (Appendix S1). Disc color characters were not selected because they showed a great amount of individual variation. *Ophiactis savignyi* was used as outgroup species (Fig. S4).

The character matrix was edited in Mesquite v.3.51 (*Maddison & Maddison, 2018*), with inapplicable data scored with "-" (Appendix S2). Parsimony analysis was conducted using TNT v.1.5 (*Goloboff & Catalano, 2016*). Optimal trees were searched using random addition sequences of Wagner trees, followed by the TBR algorithm, using 500 replicates, and saving 10 trees per replicate. The resulting trees were used as starting points for a round of TBR branch swapping. Bootstrap support values for the strict consensus tree were determined through 1,000 iterations, with default settings. Visualization, interpretation, and annotation of the cladogram were performed with FigTree, TNT, and Illustrator, respectively.

## Molecular species delimitation

Two sequence-based species delimitation methods were employed using mtDNA data only. Multi-rate Poisson tree processes (mPTP) is a non-coalescent, sequence-based, maximum likelihood method that does not require pre-determined taxonomic designations and uses statistical cutoffs to delineate taxa on a phylogenetic input tree (*Zhang et al., 2013*; *Kapli et al., 2017*). It identifies variation in the pace of branching events, modeling speciation based

on the number of substitutions. The online version of mPTP (https://mptp.h-its.org/#/tree) was used to calculate species delimitation, using the Bayesian tree as input, mPTP model selected, with nine specimens as outgroup taxa. The resulting trees were visualized using FigTree v.1.4.4.

The Bayeasian program BPP v.4.1.4 (*Yang, 2015*) was used to infer phylogenetic and phylogeographic patterns under the multispecies coalescent model (MSC) and to calculate potential species' posterior probabilities delimitation (*Yang & Rannala, 2010*; *Yang, 2015*). BPP appears to be relatively robust to the influence of unequal population sizes, rates of population growth, unbalanced sampling, and mutation rate heterogeneity (*Luo et al., 2018*), and has proven effective in analyses of taxon evolution and divergence (*Zhang et al., 2013*; *Moritz et al., 2018*). A 151-taxon dataset and the Bayesian tree were used for all BPP analyses, because sequences with missing data were eliminated. The estimation of appropriate starting species divergence times ($\tau$s) and population size parameters ($\theta$s) was initially performed through A00 BPP analysis (*Masonick & Weirauch, 2019*). This estimation was based on the expected number of mutations per kilobase, as suggested by *Yang (2015)*. The parameters for the MSC model were estimated using BPP v4.1.4 (following the A00 analysis from *Yang, 2015*). The joint species tree estimation and species delimitation analyses (A11) were carried out with inverse-gamma parameters of $\theta$ (5, 0.05) and $\tau$ (5, 0.02). The A11 analysis was run for 100,000 generations, sampling every two steps after discarding the initial 10,000 generations as burn-in. The analysis results were confirmed by conducting three independent runs for each analysis. Only lineages with a posterior probability (pp) of $\geq$ 0.95 were considered well-supported (*Masonick & Weirauch, 2019*).

## Integrative species delimitation

Morphological and mtDNA data were analyzed in a common coalescent Bayesian framework using the program iBPP v.2.1.3 (*Solís-Lemus, Knowles & Ané, 2015*). This method has been shown to improve species delimitation accuracy by incorporating molecular and quantitative phenotypic data in the assessment of *a priori* species assignments using a guide tree. iBPP analyses were performed using the Bayesian topology as a guide tree. The same values for demographic parameters $\theta$ and $\tau$ as in the BPP analysis were used. The total evidence analyses described below utilized two datasets, as iBPP is capable of incorporating morphological data represented by quantitative, continuous traits: (a) multistate character matrix with the 25 arm color-characters as trait data scored through the color pattern analysis, and (b) PC1 + PC2 values for DAP and VAP as trait data obtained through the GM analysis. For both analyses, 46 specimens were used that included sequences and GM data, and the second included the specimens with sequences and the photographic record *in vivo*. To determine if different data types result in congruent delimitations, the following comparisons were made among data types: sequence data only, coloration data only, GM data only, sequence and coloration data (iBPP$_{Seq+COL}$), and sequence and GM data (iBPP$_{Seq+GM}$) (*Masonick & Weirauch, 2019*). Posterior probabilities (pp) at each node were averaged after performing each analysis three times. After a burn-in phase of 10,000 iterations, every second tree was sampled for a total of 100,000 trees. Well-supported
delimitations were only considered for nodes on the guide tree that were recovered with pp values of ≥ 0.95 (*Masonick & Weirauch, 2019*).

## Population diversity

Standard measures of genetic diversity (number of haplotypes, haplotype diversity *h*, and nucleotide diversity $\pi$) were calculated using Arlequin v.3.5.2.2 (*Excoffier & Lischer, 2010*). Unique haplotypes were identified using DnaSP v6.12 (*Rozas et al., 2017*). Geographical relationships of mtDNA haplotypes were summarized using the TCS algorithm (*Clement et al., 2002*) in the software PopART v.1.7 (*Leigh & Bryant, 2015*). To perform the homologous character comparison, missing data were excluded by trimming sequences to 632 bp. For comparison of the extent of divergence with other ophiuroid species, evolutionary distance values were generated in MEGA 11 (*Tamura, Stecher & Kumar, 2021*) using the Kimura 2-parameter model (*Kimura, 1980*), support values based on 1,000 bootstraps, including both transitions and transversions, the rate variation among sites was modeled with a gamma distribution (shape parameter = 1), codon positions included were 1st + 2nd + 3rd, and missing data were treated as pairwise deletion (Table 1).

## Demographic history

To test for past population expansions, the neutrality Fu's *Fs* test (*Fu, 1997*) was implemented in Arlequin v.3.5.2.2, and significance was assessed with 1,000 permutations. In addition, the frequency of the distribution of mismatches was obtained in Arlequin and plotted with the R package *ggplot2* (*Wickham, 2016*) to determine whether the populations exhibit evidence of spatial/demographic expansions or a stationary population history (*Tajima, 1989*). The Raggedness index and the sum of squared deviations (SSD) obtained in Arlequin were used to analyze the goodness of fit for the population expansion model, according to *Harpending (1994)*.

# RESULTS

## Genetic differentiation and spatial distribution

The consensus COI phylogenetic trees showed three deeply divergent (K2P distances 17.0–27.9%; Table 1), highly supported (PP/bootstrap at 100/≥90) clades in *Ophiothrix angulata* with both methods (BI and ML) (Fig. 3). Two clades (Clade 2 and Clade 3) are widespread, whereas Clade 1 has a more restricted range (Fig. 1). The three clades have overlapping depth distributional ranges down to 45 m, with only Clade 3 extending deeper, with five sequenced specimens from 45–135 m (Fig. S5). COI haplotype networks recovered the same groups obtained in the phylogenetic tree. Haplogroup 1 (Clade 1) was separated from Haplogroup 2 (Clade 2) by 64 mutational steps (m-s), while Haplogroup (Clade 3) was separated by 110 m-s from Clade 1, and by 96 m-s from Clade 2. Clades 1 and 2 were not differentiated in ITS2, but Clade 3 was divergent (Fig. 4).

## Clade 1

Clade 1 (*n* = 76) was encountered only in the Gulf of Mexico and east Florida, including the Veracruz platform reefs in the Southern Gulf of Mexico and Northern Gulf of Mexico

**Table 1** Genetic distances (±standard error) between recovered clades of *O. angulata* based on a Kimura 2-Parameters model for COI. Inter-specific distance values are presented below the diagonal. Numbers along the diagonal in bold and brackets represent intra-specific variation. Genetic distances were compared with the congeneric species *Ophiothrix lineata* and *O. suensonii* distributed in the Caribbean and Gulf of Mexico; *Ophiothrix cimar* distributed in the Caribbean and *Ophiactis savignyi* (outgroup).

| COI | Clade 1 | Clade 2 | Clade 3 | *O. cimar* | *O. lineata* | *O. suensonii* | *O. savignyi* |
|---|---|---|---|---|---|---|---|
| Clade 1 | [**0.032 ± 0.004**] | | | | | | |
| Clade 2 | 0.170 ± 0.019 | [**0.007 ± 0.001**] | | | | | |
| Clade 3 | 0.279 ± 0.028 | 0.264 ± 0.026 | [**0.053 ± 0.006**] | | | | |
| *O. cimar* | 0.316 ± 0.032 | 0.310 ± 0.033 | 0.278 ± 0.028 | – | | | |
| *O. lineata* | 0.276 ± 0.028 | 0.253 ± 0.027 | 0.254 ± 0.026 | 0.133 ± 0.016 | – | | |
| *O. suensonii* | 0.304 ± 0.030 | 0.278 ± 0.029 | 0.274 ± 0.028 | 0.243 ± 0.027 | 0.264 ± 0.028 | – | |
| *O. savignyi* | 0.323 ± 0.032 | 0.302 ± 0.031 | 0.319 ± 0.032 | 0.279 ± 0.030 | 0.263 ± 0.027 | 0.302 ± 0.032 | – |

at 0–41 m depths (Fig. S5). Although it overlaps in distribution and depth with clades 2 and 3, the latter are widely distributed across the sampled areas (Fig. 1). Three sympatric subclades can be differentiated within Clade 1 (Figs. 3 and 5), that cooccur in the Northern Gulf of Mexico and the Florida Keys.

## Clade 2

Clade 2 ($n = 51$) was collected in almost all areas sampled (Northern Gulf of Mexico, Florida Keys, East Florida, Southern Gulf of Mexico, Western Caribbean, Southwestern Caribbean, and the Eastern Caribbean; Fig. 1) at 0.5–42 m depths (Fig. S5). Two deeply divergent (5.1% K2P), allopatric subclades can be differentiated in Clade 2: a widely distributed subclade 2A ($n = 49$) that displays a star-like haplotype network (Fig. 5), and subclade 2B ($n = 2$) sampled only in Guadeloupe (Eastern Caribbean).

## Clade 3

Clade 3 included 31 specimens from Northwest Florida, Florida Keys, Campeche Bank, Western Caribbean, and Eastern Caribbean (Fig. 1) at 1.5–135 m depths (Fig. S5). This clade shows high levels of differentiation, with almost all specimens having distinct COI sequences, separated by multiple substitutions up to ~5.3% K2P (Fig. 6).

## Geometric morphometrics

Procrustes ANOVA showed significant differentiation among clades for DAP and VAP ($p$-value < 0.05). Clades 1A, 1B, and 2A clustered together, while Clade 3 separated in the morpho-space and was significantly different based on pairwise comparisons among clades for both arm plates (Table S3). Corresponding thin-plate spline (TPS) deformation grids for PC1 in DAP analysis illustrate the extremes of variation with the extension/shortening of proximal and lateral edges, while PC2 shows the extension/shortening of the distal edge (Fig. 7). TPS deformation grids in VAP indicate that a considerable portion of the variance in PC1 is attributed to the extension/shortening of both, the proximal and distal edges, as well as and the lateral edges. In contrast, PC2 shows the extension/shortening of proximal and distal edges (Fig. 8). To further illustrate how clades groups in morpho-space, three-dimensional scatterplots of the three principal components (PCs) for both DAP and VAP analyses are provided (Figs. S6 and S7). Delimitations based on significantly different

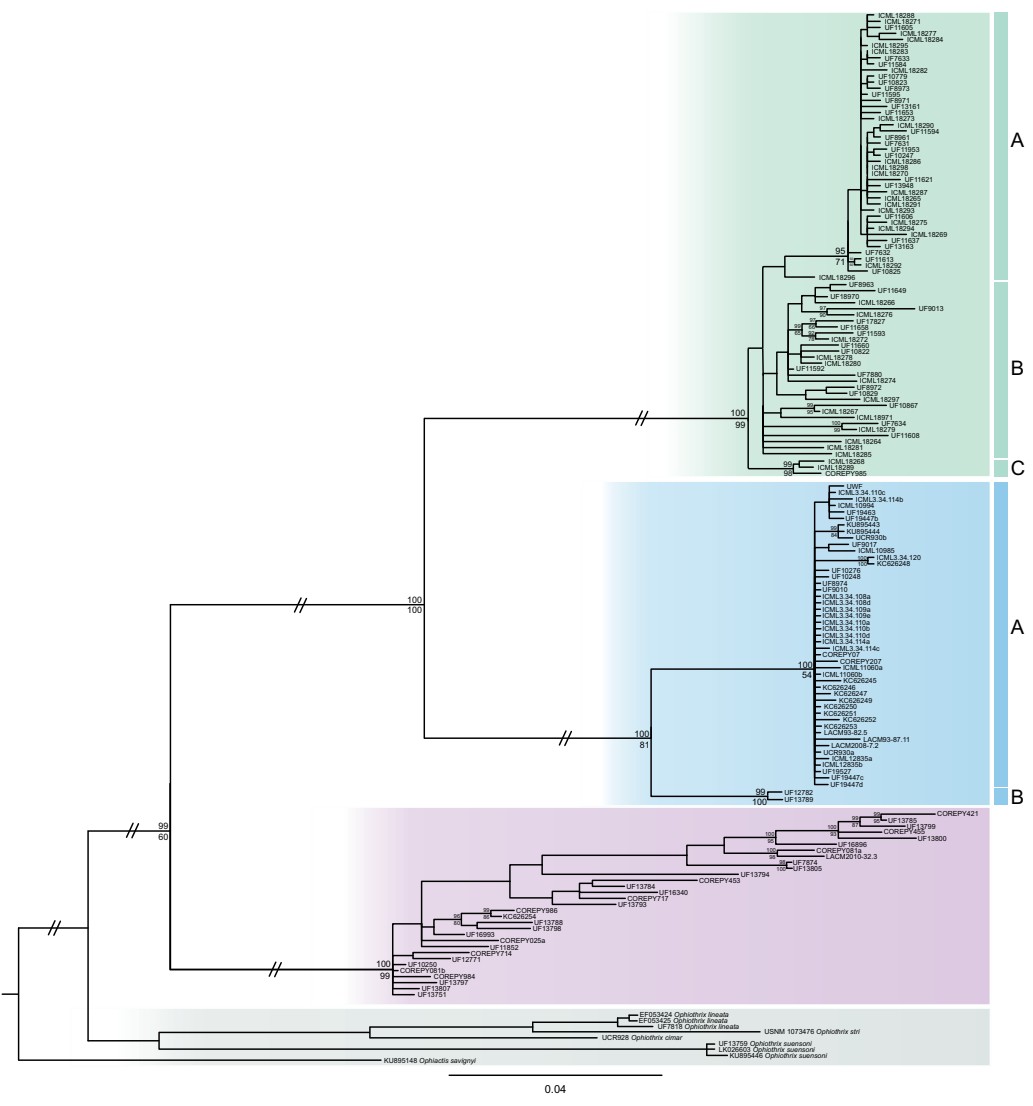

**Figure 3 MtDNA bayesian consensus tree.** Bayesian consensus tree of COI sequences produced using the GTR+I+G model in MrBayes v.3.2.7a on XSEDE for *Ophiothrix angulata* and outgroups. Clade 1 (green), Clade 2 (blue), Clade 3 (pink), and Outgroup (gray). Bayesian posterior probabilities (above), followed by ML bootstrap support (below; 1,000 replicates), are indicated at nodes.

DAP and VAP shapes tested in GM analysis are shown in Figs. 7 and 8, respectively; the DAP and VAP analysis results show a correlation with molecular evidence supporting the three clades' relationship. The Multivariate Regression showed that 16.7% of the variation for DAP and 9.0% for VAP was attributable to allometric variation in shape; this variation showed no significant interaction for any plate between Centroid size and "clade" factor in Procrustes ANOVA ($p$-value > 0.05). Measurement errors were low, 1.53% for DAP and 3.05% for VAP.

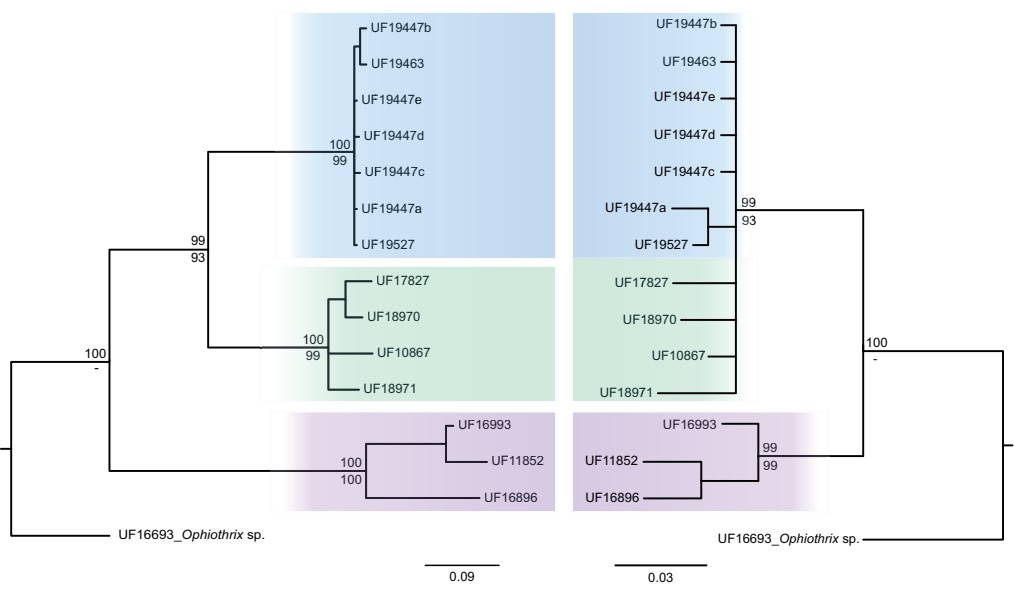

**Figure 4  MtDNA *vs* nrDNA trees.** A comparison of mtDNA (COI) (left) and nrDNA (ITS2) (right) phylogenies from the BI analysis are displayed as cladograms for *Ophiothrix angulata* with Bayesian posterior probabilities displayed above and ML bootstrap (1,000 replicates) support displayed below the nodes.

## Dorsal arm color pattern analysis

All but four (characters 16, 19, 22, and 25) of the 25 selected color-characters were parsimony informative, but none showed concordant differences with genetic clade assignment (Fig. S8). The consensus tree length was 170 steps, with CI = 0.201 and RI = 0.479. Specimens from all clades clustered together.

## Molecular species delimitation

BPP analyses recovered Clades 1, 3, and subclades 2A, and 2B as distinct, with posterior probability values of > 99 for all clades. mPTP species delimitations recovered the same clades as BPP (Fig. 9).

## Integrative species delimitation.

iBPP analyses recovered Clades 1A, 1B, 2, and 3 as distinct for both iBPP$_{Seq+GM}$ and iBPP$_{Seq}$, with high support in all runs. iBPP$_{GM}$ separated Clade 3 with high support, but low support was found between clades 1A, 1B, and 2. iBPP$_{Seq+COL}$ and iBPP$_{COL}$ did not show congruence in runs, so all clade specimens clustered together (Fig. S9).

## Population diversity

Haplotype diversity ($h$) and nucleotide diversity ($\pi$) ranged from 0.743 to 1.000 and from 0.007 to 0.048, respectively, among the clades (Table 2). TCS network recovered the same three clades as the phylogenetic analysis of mtDNA. Nucleotide ($\pi$) and haplotype ($h$) diversity values in the TCS network indicated that Clade 3 was more genetically diverse, followed by Clade 1, while Clade 2 exhibited lower nucleotide and haplotype diversity (Table 2).

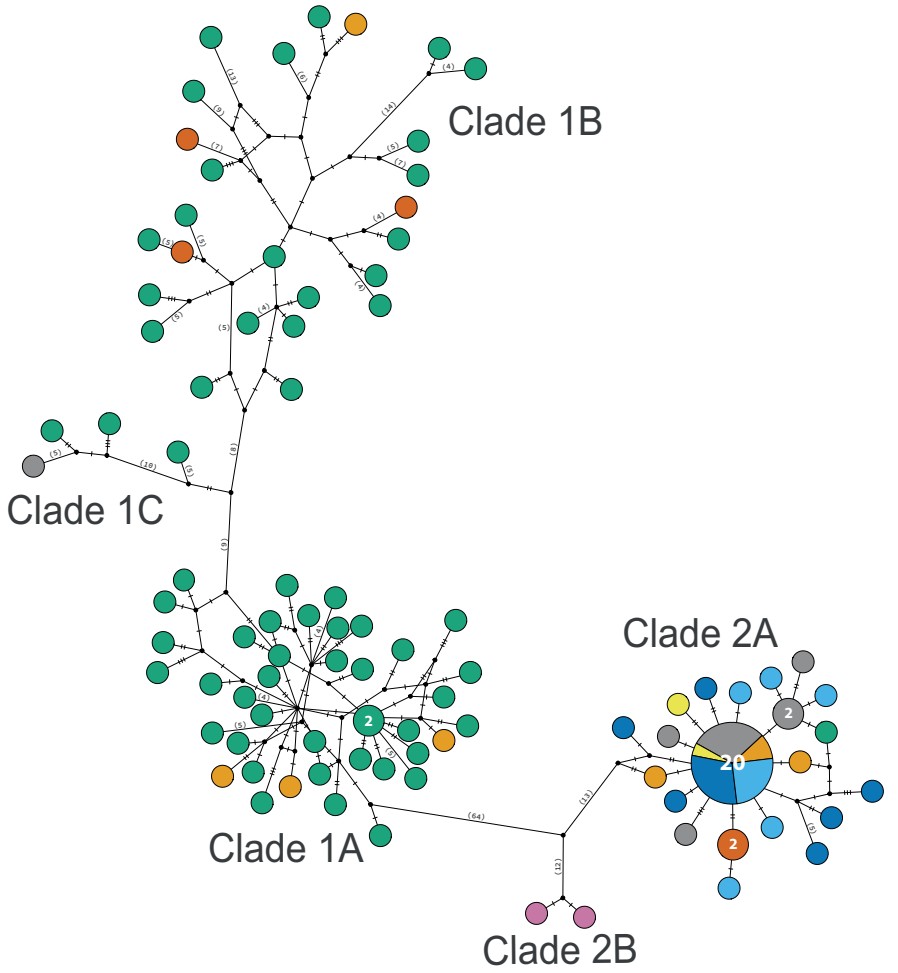

**Figure 5** **TCS haplotype network clades 1 and 2.** Clade 1A ($n = 42$), Clade 1B ($n = 26$), Clade 1C ($n = 3$), Clade 2A ($n = 42$), and Clade 2B ($n = 2$). TCS haplotype network of COI sequences for two *Ophiothrix angulata* clades (632 bp). The number of specimens is superimposed onto the more abundant haplotypes. Northern Gulf of Mexico (green). Florida Keys (orange). Eastern Florida (vermilion). Southern Gulf of Mexico (grey). Campeche Bank (yellow). Western Caribbean (blue). Southwestern Caribbean (sky blue). Eastern Caribbean (pink).

## Demographic history

Results of Fu's Fs tests, Raggedness index, and SSD analysis are provided in Table 2. Fu's neutrality test gave significant negative values for all clades, and subclades 1A and 1B, suggesting past demographic expansions. The Raggedness index and SSD were low and non-significant for all clades and subclades 1A and 1B, respectively, suggesting an unimodal distribution of mismatches as expected for a demographic expansion. The distribution of mismatches for Clades 1 and 3 was bimodal (Fig. S10), which may indicate demographic balance. However, when clades 1A and 1B were analyzed separately, mismatches showed a unimodal distribution for 1A while continued bimodal for 1B (Fig. S11). The mismatch distribution of Clade 2 was unimodal (Fig. S10).

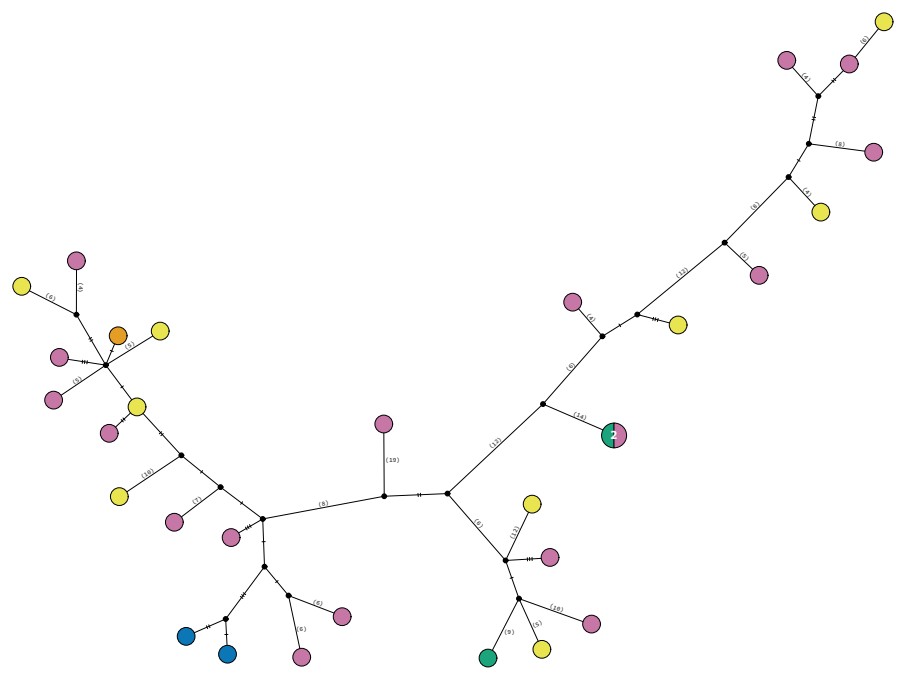

**Figure 6   TCS haplotype network Clade 3 ($n = 31$).** TCS haplotype network of COI sequences for one *Ophiothrix angulata* clade (632 pb). The number of specimens is superimposed onto the more abundant haplotypes. Northern Gulf of Mexico (green). Florida Keys (orange). Campeche Bank (yellow). Western Caribbean (blue). Eastern Caribbean (pink).

## DISCUSSION

### Species delimitation in *Ophiothrix angulata*

Analysis of COI sequence data revealed three deeply-divergent clades within *Ophiothrix angulata* in the tropical Western Atlantic, one of which (Clade 3) was also divergent in ITS2 and geometric morphometrics. Species delineation algorithms recovered the same three clades using genetic and combined genetic and morphometric data. The deep level of genetic differentiation with COI among these clades is also consistent with the three lineages representing separate species (Table 1).

The consistency between morphological and genetic data on differentiating Clade 3, together with its co-occurrence with clades 1 and 2, demonstrates the lack of gene flow between Clade 3 and the others, and clearly establishes that it is a separate biological species; thus, it can be considered a confirmed candidate species (CCS) (*Padial et al., 2010*). All three clades co-occur in the Florida Keys, where they were collected on the same day, site, depth, and habitat (UF10247-1A, UF10248-2A, and UF10250-3 in Table S1), which further suggests reproductive isolation for Clade 3.

The status of clades 1 and 2 remain open, as they were separated only by COI sequences, and thus could be species or deep conspecific lineages (DCL) (*Padial et al., 2010*). While the distribution of these two clades overlaps and they are sympatric at several localities surveyed, Clade 1 is mostly known from around central and North Florida, while Clade

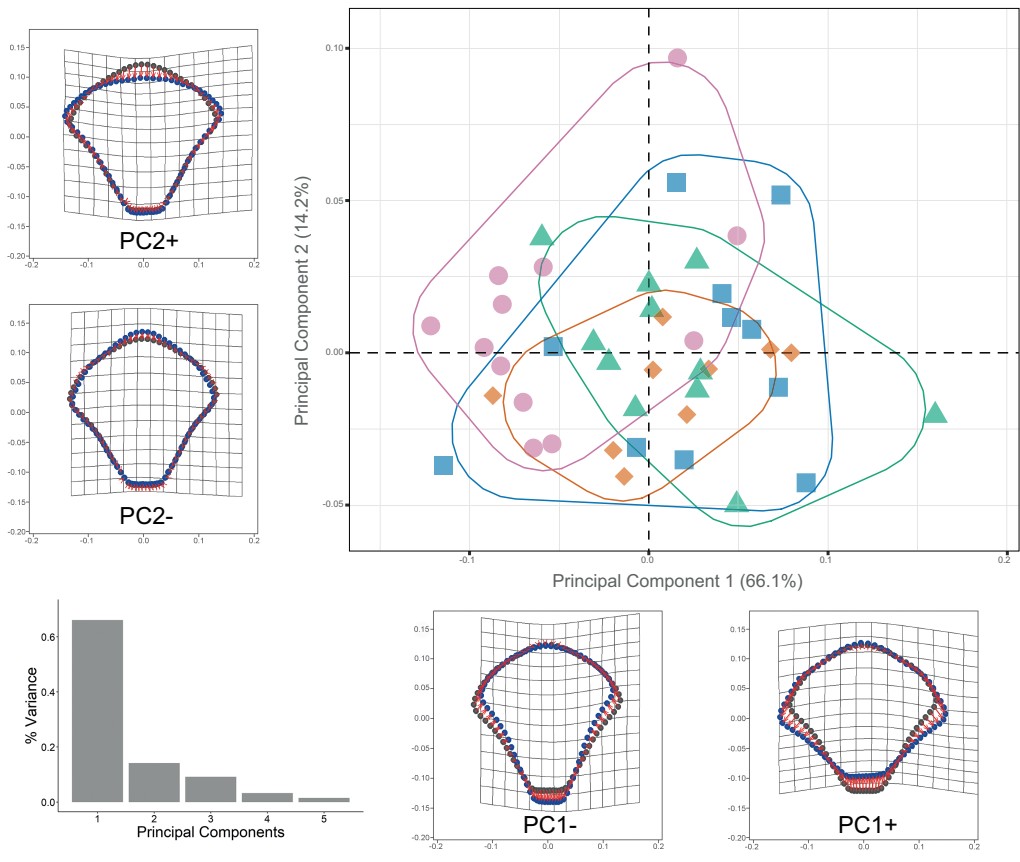

**Figure 7** **Dorsal arm plate geometric morphometrics.** Scatterplots showing shape variation along principal component axes. Clade 1A (green). Clade 1B (orange). Clade 2 (blue). Clade 3 (pink). Thin-plate spline deformation grids accompany each PC axis to show the specimens' shape at their positive and negative ends; the arm plate consensus shape is gray. The bar graph depicts the percentage of variance explained by PC axes.

2 was encountered in South Florida, Southern Gulf of Mexico, and the Caribbean. This distribution largely reflects the differentiation of Gatunian and Callosahatchian faunas that have been established since the Miocene (*Vermeij, 2005*) and suggest allopatric differentiation along this boundary. Additional studies are needed to assess their status.

Clustering sequences into three subclades in Clade 1 and two subclades in Clade 2 is more challenging to interpret. In Clade 2, the two subclades are allopatric, with one represented by two specimens from the Lesser Antilles, the other widespread, suggesting geographic differentiation.

## Geometric morphometrics and Dorsal arm color pattern

The shape of both the dorsal and ventral arm plates proved to be a useful indicator for distinguishing Clade 3 from Clades 1-2. Geometric morphometrics has been employed in echinoderm studies to investigate different approaches to understanding the biology and classification of the different orders and families. For instance, *Martínez-Melo, De Luna & Buitrón-Sánchez (2017)* used GM methods to differentiate between genera in the

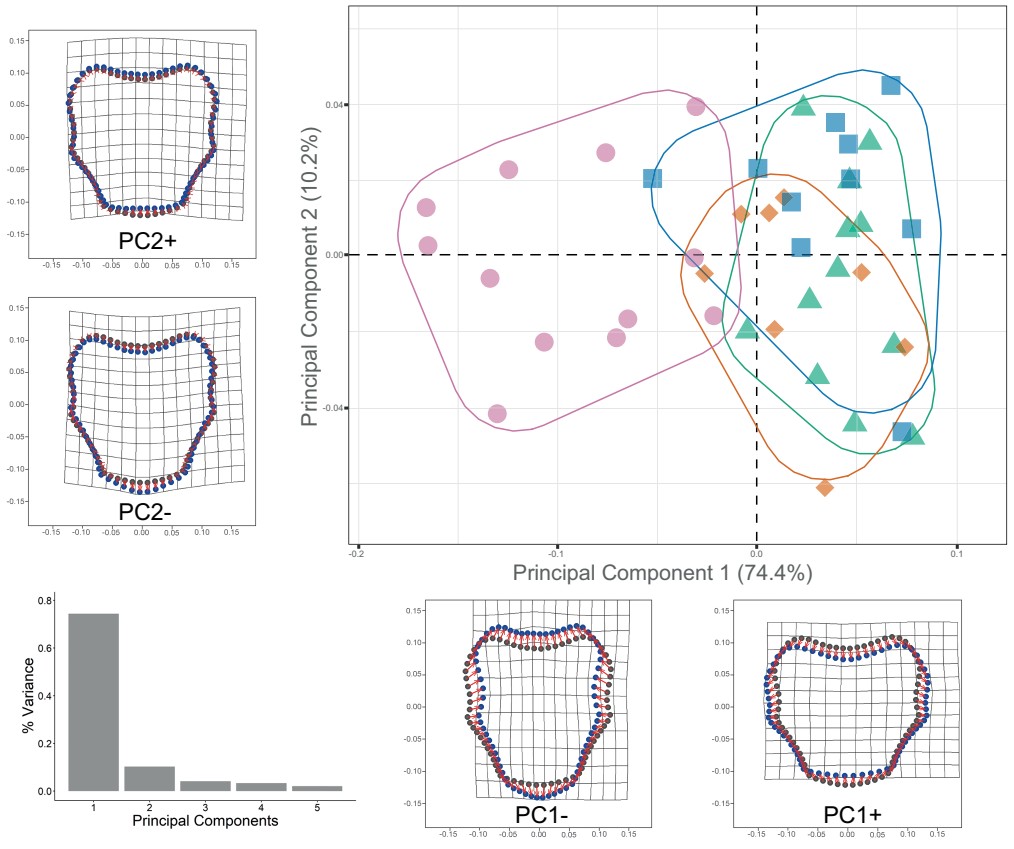

**Figure 8** **Ventral arm plate geometric morphometrics.** Scatterplots showing shape variation along principal component axes. Clade 1A (green). Clade 1B (orange). Clade 2 (blue). Clade 3 (pink). Thin-plate spline deformation grids accompany each PC axis to show the specimens' shape at their positive and negative ends; the arm plate consensus shape is gray. The bar graph depicts the percentage of variance explained by PC axes.

Cassidulidae family based on the cryptic morphology of plate shapes in Echinoidea. *De los Palos-Peña et al. (2021)*, combined scanning electron microscopy and ontogenetic studies of the odontophore in *Luidia superba* to understand patterns of size and shape variation. Similarly, *Swisher (2021)* studied ontogeny in fossil clypeasteroids and confirmed size and shape changes in the oral/aboral plates using GM methods. However, our study incorporates the concept of allometry, which considers the effect of size on shape variation due to ontogeny and other ecological factors influencing morphology (*Benítez et al., 2013*; *Klingenberg, 2022*).

Closely related species frequently differ in color pattern and color differences are often among the first visible morphological changes that appear among differentiating species (*Benavides-Serrato & O'Hara, 2008*; *Hoareau et al., 2013*). *Ophiothrix angulata* displays high polymorphism in color patterns (Figs. S1, S2, and S3), and color differences have been suggested to potentially reflect cryptic species differentiation in this species (*Clark, 1918*; *Tommasi, 1970*). The absence of correlation between color pattern and genetic clade

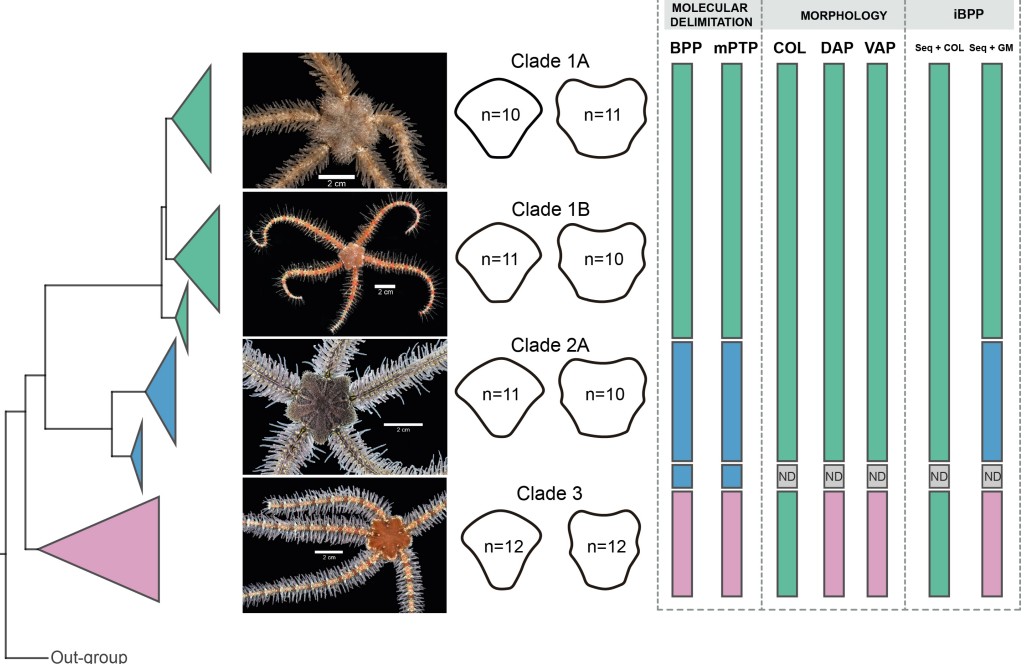

**Figure 9** **Species delimitation for *Ophiothrix angulata* clades based on molecular, morphological, and integrative approaches.** Clade 1A (UF7632), Clade 1B (UF11608), Clade 2A (UF10248), and Clade 3 (UF10250). COI phylogeny on the left, representative specimen based on DAP and VAP consensus shape of major clades in the middle, species delineations on the right. Delineated species are represented by separate colors. iBPP results are based on (A) COI and arm color information (Seq+COL) and (B) COI and geometric morphometric data (Seq+GM). ND, no data. Photo credit: Invertebrate Zoology Collection, Florida Museum of Natural History, University of Florida.

**Table 2** **Summary statistics and demographic analyses for the largest clades of *O. angulata*;** N = number individuals, Number of haplotypes, h = haplotype diversity, $\pi$ = nucleotide diversity ± standard deviation, SSD = Sum of squared differences in mismatch analysis, Mismatch distribution raggedness index ($r$), results of Fu's $Fs$. Clade 1C was not considered because its $n = 3$.

| Clades | N | Number of haplotypes | h | $\pi$ | SSD | | Raggedness $r$ | | Fu's $Fs$ | |
|---|---|---|---|---|---|---|---|---|---|---|
| | | | | | SSD | $p$-value | $r$ | $p$-value | $Fs$ | $p$-value |
| Clade 1 | 76 | 74 | 0.999 ± 0.003 | 0.032 ± 0.001 | 0.023 | 0.129 | 0.005 | 0.320 | −72.71 | 0.000 |
| Clade 1A | 44 | 43 | 0.998 ± 0.005 | 0.010 ± 0.001 | 0.004 | 0.143 | 0.018 | 0.156 | −34.17 | 0.000 |
| Clade 1B | 29 | 29 | 1.000 ± 0.009 | 0.026 ± 0.002 | 0.013 | 0.187 | 0.007 | 0.889 | −18.64 | 0.000 |
| Clade 2 | 51 | 23 | 0.743 ± 0.068 | 0.007 ± 0.002 | 0.006 | 0.810 | 0.015 | 0.972 | −11.73 | 0.000 |
| Clade 3 | 31 | 30 | 0.998 ± 0.009 | 0.048 ± 0.003 | 0.004 | 0.914 | 0.009 | 0.574 | −08.73 | 0.007 |

in our study suggests that color variation is an intra-specific trait. This finding is consistent with previous studies on the *Ophiothrix fragilis* complex, a widely distributed species in the Northeastern Atlantic Ocean. Previous attempts to link genetic lineages (*Baric & Sturmbauer, 1999*; *Muths et al., 2009*; *Taboada & Pérez-Portela, 2016*, for lineages 1 and 2) with some of the color variants identified by *Koehler (1921)* were unsuccessful.

## COI divergence in ophiuroids

COI K2P distances among the three clades (17.0–27.9%) were higher than the mean inter-specific divergence among most ophiuroid species. In a DNA barcoding study of 503 specimens of 191 ophiuroid species, *Ward, Holmes & O'Hara (2008)* found intra-specific variation to range 0–3% (mean = 0.62%), whereas inter-specific divergence within genera averaged 15%. High levels of genetic divergence encountered in other ophiuroid species have generally led to the recognition of multiple cryptic species. Examples include the *Ophiothrix fragilis* complex: K2P distance = 18.6% (*Muths et al., 2009*), 15–17% (*Pérez-Portela, Almada & Turon, 2013*), 19–22% (*Taboada & Pérez-Portela, 2016*); *Ophioderma longicaudum* complex: K2P distance = 2.2–10.2% (*Boissin, Stöhr & Chenuil, 2011*), 0.8–10.7% (*Weber, Stöhr & Chenuil, 2019*); *Ophiomyxa vivipara*, *Ophiacantha vivipara*, *Ophiura ooplax*, *Ophiactis abyssicola* and *Ophiothrix aristulata* complexes: K2P distance = 2.9–3.7%, 14.1–16.7%, 22%, 6.7%, and 22.9%, respectively (*O'Hara et al., 2014a*); and *Ophiacantha wolfarntzi* complex: K2P distance = 5.4–25.7% (*Martín-Ledo, Sands & López-González, 2013*). The differentiation among Clades 1, 2, and 3 in COI is thus in line with species-level differences in other ophiuroids.

## Haplotype diversity and demographic history

Haplotype networks display contrasting topologies, with Clades 1 and 3 showing higher intra-specific diversity than Clade 2. Clade 1 shows a great diversity of haplotypes in the Northern Gulf of Mexico off Florida, where all three subclades were present. In contrast, specimens from Eastern Florida, the Florida Keys, and the Southern Gulf of Mexico each fell into single and different subclades. This pattern is suggestive of allopatric differentiation along the periphery of the Clade's range, but few samples were available from these areas making interpretation challenging. Historical demography results show evidence of recent expansion (Fu's, r, SSD, and mismatches) for subclade 1A.

Clade 2 showed a star-like network suggesting recent population expansion (*Allcock & Strugnell, 2012*). Most haplotypes were within 3 m-s of the common one, except for the two specimens from Guadeloupe that were 30 m-s distant. These results suggest that continental populations along North and Central America are isolated from the insular population sampled in the Lesser Antilles, and that the former, at least, underwent a recent expansion, potentially following the Last Glacial Maximum.

Clade 3 showed the highest haplotype diversity and did not display signs of recent expansion in all analyses but showed a high diversity of haplotypes. It is noteworthy that this is the only clade that was represented among deeper water samples, suggesting that it may have the most extensive depth range, and thus potentially greater physiological tolerance. The high gene diversity has resulted in a larger population size than the other clades, suggesting success in exploitation and colonization of habitats. Some haplotypes in this clade were shared between localities separated by more than 500 km.

## CONCLUSIONS

This study provides a broad evaluation of the systematics of one of the most common ophiuroids in the Tropical Western Atlantic, *Ophiothrix angulata*. COI sequence data

revealed three deeply divergent genetic lineages. The high color variability exhibited by this group did not correlate with lineages suggesting that it represents intra-specific polymorphism. Clade 3 was separated by mtDNA, nrDNA, molecular species delimitation, the shape of dorsal and ventral arm plates, and the integrative analysis with mtDNA and geometric morphometric data. Therefore, we consider it as a confirmed candidate species. Results demonstrate that a thorough arm morphology analysis can help differentiate clades within this species complex. Molecular analyses and *in situ* records show that all three clades co-occur in some areas. For Clades 1A and 2, Fu's, r, SSD, and mismatches showed evidence of recent expansion. Additional geographic sampling combined with physiological, reproductive, and ecological data incorporating phylogeographic analysis may further resolve this species complex.

## ACKNOWLEDGEMENTS

We thank David Pawson, Christopher Pomory, Gordon Hendler, Juan José Alvarado, and Nuno Simões for their contribution with samples for sequencing. We thank Tania Pineda, Daniel Janies, and Antar Pérez for their help in sequencing extra material; Tania Pineda, Carolina Martín, Andrea Caballero, and Tayra Parada for images of collection labels; Berenit Mendoza (Laboratorio de Microscopía Electrónica, IB-UNAM) for their technical support during the acquisition of SEM images, Susana Guzmán (Laboratorio de Microscopía y Fotografía de la Biodiversidad II, IB-UNAM) and Miguel Pérez (Laboratorio de Cómputo de Alto Rendimiento, FC-UNAM) for their technical support; Antar Pérez (PopArt, BPP), Sandra Ospina (*geomorph*), Jesús Díaz (TNT), Maite Mascaró and Dorottya Angyal for their constructive data analysis advice. We thank to the museum curators and collection managers for facilitating access to their collections Amanda Bemis, John Slapcinsky, Adam Baldinger, Penny Benson, Chad Walter, Alicia Durán, Pedro Homa, Raúl Castillo, and Leonardo Chacón.

### Funding

This work was supported by CONACYT scholarship CVU 271645 through the Posgrado de Ciencias del Mar y Limnología, UNAM, and the Ernst Mayr Grant of the Museum of Comparative Zoology of Harvard University. Sequencing costs were supported by NSF DEB 0529724. The funders had no role in study design, data collection and analysis, decision to publish, or preparation of the manuscript.

### Grant Disclosures

The following grant information was disclosed by the authors:
CONACYT scholarship CVU 271645 through the Posgrado de Ciencias del Mar y Limnología, UNAM.
The Ernst Mayr Grant of the Museum of Comparative Zoology of Harvard University.
NSF DEB 0529724.

## Competing Interests

The authors declare that there are no competing interests.

## Author Contributions

- Yoalli Quetzalli Hernández-Díaz conceived and designed the experiments, performed the experiments, analyzed the data, prepared figures and/or tables, authored or reviewed drafts of the article, and approved the final draft.
- Francisco Solis conceived and designed the experiments, authored or reviewed drafts of the article, and approved the final draft.
- Rosa G. Beltrán-López analyzed the data, authored or reviewed drafts of the article, and approved the final draft.
- Hugo A. Benítez analyzed the data, authored or reviewed drafts of the article, and approved the final draft.
- Píndaro Díaz-Jaimes analyzed the data, authored or reviewed drafts of the article, and approved the final draft.
- Gustav Paulay analyzed the data, authored or reviewed drafts of the article, and approved the final draft.

## Field Study Permissions

The following information was supplied relating to field study approvals (i.e., approving body and any reference numbers):

Field sample collection where approved by Secretaria de Agricultura, Ganadería, Desarrollo Rural, Pesca y Alimentación (SAGARPA).

## DNA Deposition

The following information was supplied regarding the deposition of DNA sequences:

The mtDNA COI sequences are available at GenBank: COI: MT338285 to MT338398, and ON245084 to ON245096.

Sequences ITS2 are available at GenBank: OQ225473 to OQ225482.

## Data Deposition

The raw data employed for Parsimony analysis, for depths and frequency and for geometric morphometrics analysis are in the Supplemental Files.

## Supplemental Information

Supplemental information for this article can be found online at http://dx.doi.org/10.7717/peerj.15655#supplemental-information.

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
