# Peer review of "Integrative species delimitation in the common ophiuroid Ophiothrix angulata (Echinodermata: Ophiuroidea): insights from COI, ITS2, arm coloration, and geometric morphometrics"

_PeerJ, doi:10.7717/peerj.15655_

## Round 0.1 · original submission · Minor Revisions

Dear authors,

After reviewing the article entitled "Integrative species delimitation in the common ophiuroid Ophiothrix angulata (Echinodermata: Ophiuroidea): insights from COI, ITS2, arm coloration, and geometric morphometrics" and receiving the comments of three reviewers, it is necessary to make some minor corrections, Mainly the reviewers comment that it is necessary to review the English, review the citation format used by PeerJ and finally review the figures and append some figures that are in the supplementary material to the main text. I am sure that these suggestions will help to improve this interesting manuscript that integrates molecular data and morphometry to delimit species.

Sincerely,

Armando Sunny

Reviewer 1 ·

Basic reporting

English is unclear and failed many times. I recommend revising with a native with knowledge in the area.

The references are old in many sentences. In this case, is recommendable to add more recent references; also, some sentences lack references, and one has an inadequate format.

In addition, I couldn't observe the results of the geometric morphometrics analysis in Figures 7 and 8. I don't know if was a failure of the PDF build, but the figures didn't show anything.

Experimental design

It is an interesting study and it is well-directed, it complies with answering the research question. However, I would recommend improving the introduction, is weak and in some parts, repetitive. The background part seems a bit weak to me, I don't know if there are more and more recent studies, but expanding this part could adequately support the hypothesis and objective of the work.

There are some format details to correct in methods, please review them.

Validity of the findings

No comment

Annotated reviews are not available for download in order to protect the identity of reviewers who chose to remain anonymous.

·

Basic reporting

This is a very nice paper integrating some molecular and morphological data to distinguish species level clades in the important west Atlantic brittle-star species Ophiothrix angulata. It is novel, well written, good English with a nice range of figures. I have only minor editorial comments on the ms.

Experimental design

good

Validity of the findings

valid

Additional comments

Minor editorial comments:
1) line 83, should be 'attributed' not 'attribute'.
2) lines 216-223. State whether this analysis uses the COI-only or COI-ITS2 tree.
3) Figure references. I noticed that the figure citations in the text do not match the Peerj allocated figure number (after fig 3). This is presumably because you have uploaded 3A,B & C and 6A & B as separate figures and they have been allocated separate figure numbers. Something to be aware of in the final proof.
4) line 329. Change 'no one' to 'none'
5) Lines 390-1. This is also true of the O. fragilis complex, with colour/disc spine forms not matching genetic clades. Numerous 'varieties' have also been described for fragilis, e.g. Koehler 1921 'Fauna de France', although the genetic lineage III reported by Taboata & Pérez-Portela, 2016 is actually O. luetkeni.
6) Line 422. should be 'display' not displayed'

Reviewer 3 ·

Basic reporting

No comment.

Experimental design

No comment.

Validity of the findings

No comment.

Additional comments

See attached PDF.

Annotated reviews are not available for download in order to protect the identity of reviewers who chose to remain anonymous.

---

## Round 0.2 · Minor Revisions

Dear Authors,

I am pleased to inform you that your manuscript is nearing acceptance, with two reviewers having already given their approval. However, there are some minor suggestions from Reviewers 2 & 3 that need to be addressed before final acceptance. Therefore, we request that you make the necessary minor corrections to the manuscript.

Thank you for your hard work and dedication to this project. We look forward to receiving the updated manuscript.

Best regards,

Armando Sunny

Reviewer 1 ·

Basic reporting

No comment

Experimental design

No comment

Validity of the findings

No comment

Additional comments

The authors attended to my comments and carefully made the changes appropriately, so I consider that the manuscript is ready for publication.

·

Basic reporting

This paper is almost ready for publication but a few more details need to be addressed.

Lines 130-144. The methods still do not indicate which genes are included in the final Bayesian (Fig. 3) and RAxML-HPC2 trees. I presume it is just COI, but equally you may have used a supertree approach and included some ITS2 as well. Can you please make this clear.

Relatedly, the captions for figures 3 and 4 are inadequate. You need to explicitly indicate which gene is analysed for Figure 3. Also why figure the Bayesian tree, when it is the RAxML-HPC2 topology that is used in the subsequent analyses? For Fig 4, you need to indicate which of the Bayesian and ML bootstrap values are above and below the node.

Experimental design

OK

Validity of the findings

OK

Additional comments

OK

Reviewer 3 ·

Basic reporting

The new document presents remarkable improvements. The ideas, images and supplements are better organized.

Experimental design

No comment

Validity of the findings

No comment

Additional comments

No comment

Annotated reviews are not available for download in order to protect the identity of reviewers who chose to remain anonymous.

---

## Round 0.3 · accepted · Accept

Dear Authors,

I am delighted to announce that, upon reviewing the revisions made, your manuscript has been accepted for publication in PeerJ. I would like to express my sincere appreciation for your patience and dedication in addressing the feedback provided.

Once again, congratulations on this accomplishment, and I look forward to seeing your work published in Peerj.

Best regards,

Armando Sunny